# Ginger: An Efficient Curvature Approximation with Linear Complexity for General Neural Networks

## Abstract

Second-order optimization approaches like the generalized Gauss-Newton method are considered more powerful as they utilize the curvature information of the objective function with preconditioning matrices. Albeit offering tempting theoretical benefits, they are not easily applicable to modern deep learning. The major reason is due to the quadratic memory and cubic time complexity to compute the inverse of the matrix. These requirements are infeasible even with state-of-the-art hardware. In this work, we propose **Ginger**, an ei**g**endecomposition for the **i**nverse of the ge**n**eralized **G**auss-N**e**wton mat**r**ix. Our method enjoys efficient linear memory and time complexity for each iteration. Instead of approximating the conditioning matrix, we directly maintain its inverse to make the approximation more accurate. We provide the convergence result of GINGER for non-convex objectives. Our experiments on different tasks with different model architectures verify the effectiveness of our method.

## 1 Introduction

Second-order optimization methods are usually more powerful by considering the curvature information of the objective function with preconditioning matrices. However, such methods are impractical due to the prohibitive memory and time cost. Specifically, for a neural network with $d$ parameters, the full-matrix preconditioning requires quadratic memory to store and cubic time for inverse at each iteration. Consider a Transformer (Vaswani et al., 2017) model with 80M parameters, the full-matrix preconditioning requires 3 petabytes memory solely to store the matrix, not to mention the computation time to obtain its inverse.

In practice, deep learning models are usually trained with the diagonal approximation of such matrices, such as AdaGrad (Duchi et al., 2011) and its variants (Hinton et al., 2012; Kingma & Ba, 2015; Liu et al., 2023). These methods only require linear memory and linear time complexity by using the element-wise inverse of the preconditioning matrix. However, the diagonal approximation over-simplifies the curvature information because it ignores the off-diagonal elements that contain the correlation between parameters.

There are numerous attempts to approximate the full-matrix preconditioning with affordable memory and time complexity. For instance, K-FAC (Martens & Grosse, 2015; Grosse & Martens, 2016; Martens et al., 2018) uses the Kronecker-factored approximation to reconstruct the preconditioning matrix. However, such approximation is limited to specific model architectures like feed-forward neural networks (FFNs) or convolutional neural networks (CNNs). More importantly, the complexity is super-linear in the model size, making it impractical to nowadays large models. Recently, He et al. (2022) proposed a quasi-natural gradient (QNG) method that approximates the full-matrix preconditioning by factorizing it into the product of multiple simple matrices. This approximation allows the QNG method to achieve linear memory and time complexity. However, we discuss in Observation 1 that this approximation tends to be inaccurate, leading to a worse approximation.

In this work, we propose GINGER, a new derivation to approximate the preconditioning matrix without factorization. GINGER enjoys the same linear memory and time complexity as QNG, but with a more accurate approximation. We provide the convergence result of GINGER for non-convex objectives. Empirically, we show the effectiveness of GINGER across different tasks and model architectures.

## 2 Approach

### 2.1 Background: generalized Gauss–Newton and natural gradient methods

In the context of machine learning, we usually model a conditional distribution by defining

$$p_\theta(y|x) := r(y|f(\theta; x)), \tag{1}$$

where $f$ is a function on the input $x$ with some model parameter $\theta$, and $r(y|z)$ is a distribution in the exponential family (e.g., softmax). The parameter $\theta \in \mathbb{R}^d$ is trained by maximum likelihood estimation (MLE), for a given dataset $\mathcal{D} = \{(x_i, y_i)\}_{i=1}^m$. This is equivalent to minimizing the negative log-likelihood:

$$\mathcal{L}(\theta) := \frac{1}{|\mathcal{D}|} \sum_{(x,y) \in \mathcal{D}} \mathcal{L}(\theta; x, y) = \frac{1}{|\mathcal{D}|} \sum_{(x,y) \in \mathcal{D}} [-\log p_\theta(y|x)], \tag{2}$$

where $\mathcal{L}(\theta)$ and $\mathcal{L}(\theta; x, y)$ are the losses for the whole dataset and for a sample, respectively.

Second-order optimization methods Nocedal & Wright (1999) are appealing for solving the optimization problem above because they often enjoy faster convergence by utilizing the curvature information. Specifically, Newton's method, a well-known second-order approach, updates the parameters with the following rule:

$$\theta_{t+1} \leftarrow \theta_t - \eta_t \left(\nabla^2 \mathcal{L}(\theta_t)\right)^{-1} \nabla \mathcal{L}(\theta_t), \tag{3}$$

where $\eta_t > 0$ is the learning rate and $\nabla^2 \mathcal{L}(\theta_t)$ is the second-order derivative, known as the Hessian matrix.

**The generalized Gauss–Newton method.** The standard Newton's method may not work well for non-convex optimization, because the preconditioning matrix may not be positive semi-definite. Ortega & Rheinboldt (2000) show that the Hessian matrix can be decomposed as

$$\nabla^2 \mathcal{L}(\theta) = \frac{1}{|\mathcal{D}|} \sum_{(x,y) \in \mathcal{D}} \left[ \frac{\partial f(\theta; x)}{\partial \theta}^\top \frac{\partial^2 \mathcal{L}(\theta; x, y)}{\partial f(\theta; x)^2} \frac{\partial f(\theta; x)}{\partial \theta} + \sum_{i=1}^c \frac{\partial^2 f^{(i)}(\theta; x)}{\partial \theta^2} \frac{\partial \mathcal{L}(\theta; x, y)}{\partial f^{(i)}(\theta; x)} \right],$$

where $\frac{\partial f(\theta)}{\partial \theta}$ is the Jacobian matrix of $f(\theta)$, and $f^{(i)}(\theta)$ refers to the $i$th element of the function $f(\theta)$ in (1).

In practice, the second term inside the summation is found to be less important than the first one (Sankar et al., 2021). This finding results in the following biased approximation of the Hessian matrix by

$$G := \frac{1}{|\mathcal{D}|} \sum_{(x,y) \in \mathcal{D}} \frac{\partial f(\theta; x)}{\partial \theta}^\top \frac{\partial^2 \mathcal{L}(\theta; x, y)}{\partial f(\theta; x)^2} \frac{\partial f(\theta; x)}{\partial \theta}, \tag{4}$$

where $G$ is named the generalized Gauss–Newton (GGN) matrix (Ortega & Rheinboldt, 2000; Schraudolph, 2002).

**The connection to natural gradient.** In our settings where $r(y|z)$ is in the exponential family, the matrix in the middle can be rewritten as

$$\frac{\partial^2 \mathcal{L}(\theta; x, y)}{\partial f(\theta; x)^2} = \mathbb{E}_{\hat{y} \sim r(\cdot|f(\theta; x))} \left[ \frac{\partial \log r(\hat{y}|f(\theta; x))}{\partial f(\theta; x)} \frac{\partial \log r(\hat{y}|f(\theta; x))}{\partial f(\theta; x)}^\top \right], \tag{5}$$

which is a matrix independent of the target label $y$. Putting (5) into (4), we have

$$G = \frac{1}{|\mathcal{D}|} \sum_{(x,\cdot) \in \mathcal{D}} \mathbb{E}_{\hat{y} \sim p_\theta(\cdot|x)} \left[ \nabla_\theta \log p_\theta(\hat{y}|x) \nabla_\theta \log p_\theta(\hat{y}|x)^\top \right]. \tag{6}$$

The last equation is commonly named the *Fisher information matrix* (Fisher, 1920) in the context of machine learning. This connection has been established in previous literature (Martens, 2020). It reveals a simple way to approximate the Hessian matrix by solely using the first-order derivatives when $r(y|f(\theta; x))$ is in the exponential family. This condition actually holds in many important applications, such as language models (Vaswani et al., 2017), image classifiers (He et al., 2016), and diffusion models (Ho et al., 2020). We hence leverage the connection and only consider the form of $G$ given by Equation (6) in this paper.

**Stochastic natural gradient descent.** The computation of the exact GGN matrix $G$ is usually not feasible because the dataset may contain an enormous number of data points. A remedy to this is maintaining an exponential moving average of $G_t$ at iteration step $t \in \mathbb{N}$. This can be computed iteratively using the following rule:

$$G_t \leftarrow \alpha G_{t-1} + (1 - \alpha) d_t d_t^\top, \tag{7}$$

where for simplicity we define

$$d_t := \frac{1}{\sqrt{|\mathcal{B}_t|}} \sum_{(x,\cdot) \in \mathcal{B}_t} \nabla_{\theta_t} \log p_{\theta_t}(\hat{y}|x). \tag{8}$$

Here, $\mathcal{B}_t$ is a mini-batch independently sampled to the training data, and $\hat{y}$ is a sampled prediction drawn from $p_{\theta_t}$ for each input $x$. The decay rate $\alpha \in (0, 1)$ controls the strength of the moving average, and $G_0$ is an initialization usually set as an identity matrix. If $\theta_t$ is fixed, it is easy to verify that this estimation in expectation converges to the exact $G$ when $t \to \infty$.

Although this seems to solve the problem of large datasets, the memory complexity for storing $G_t$ is at least $O(d^2)$. Even worse, the pseudoinverse of $G_t$ takes $O(d^3)$ time. Preferably, both time and memory complexities should not exceed linear to make the GGN method feasible for modern large neural networks.

## 2.2 Quasi-natural gradient method

Recently, He et al. (2022) proposed a novel method, called quasi-natural gradient (QNG), that constructs the GGN matrix in linear space and time. The procedure first factorizes $G_t = A_{t+1} A_{t+1}^\top$, since $G_t$ should always be positive semi-definite (PSD). The rule for updating $A_t$ is given by:

$$A_t = A_1 K_1 K_2 \dots K_{t-1} = \underbrace{(\sqrt{\alpha} I + \beta_1 q_1 q_1^\top)}_{=:K_1} \dots \underbrace{(\sqrt{\alpha} I + \beta_{t-1} q_{t-1} q_{t-1}^\top)}_{=:K_{t-1}}, \tag{9}$$

where $A_1$ is again set to identity, and we define $q_t := A_t^{-1} d_t$ and $\beta_t = \frac{1}{\|q_t\|^2}(\sqrt{\alpha + (1-\alpha)\|q_t\|^2} - \sqrt{\alpha})$. It is then easy to show that

$$G_t = A_t K_t K_t^\top A_t^\top = \alpha G_{t-1} + (1 - \alpha) d_t d_t^\top, \tag{10}$$

which recovers the form of the exponential moving average defined in Equation (7).

However, it is impossible to store all $K_t$ matrices from the beginning. Thus, QNG intuitively maintains the last $\tau$ matrices and estimate each $A_t$ as

$$A_t \approx \hat{A}_t := A_1 \hat{K}_{t-\tau} \hat{K}_{t-\tau+1} \dots \hat{K}_{t-1} = \underbrace{(\sqrt{\alpha} I + \beta_{t-\tau} \hat{q}_{t-\tau} \hat{q}_{t-\tau}^\top)}_{=:\hat{K}_{t-\tau}} \dots \underbrace{(\sqrt{\alpha} I + \beta_{t-1} \hat{q}_{t-1} \hat{q}_{t-1}^\top)}_{=:\hat{K}_{t-1}}, \tag{11}$$

where $\hat{K}_t$ depends on $\hat{q}_t$, which in turn depends on truncated $\hat{A}_t$ by $\hat{q}_t := \hat{A}_t^{-1} d_t$.

Given the derivation above, we state our observation as follows.

**Observation 1.** *The QNG in He et al. (2022) essentially approximates the GGN matrix with the form $G_t = \alpha^{\min(\tau,t)} I + Q_t Q_t^\top$ for some $Q_t \in \mathbb{R}^{d \times 2\min(\tau,t)}$.*

This can be seen by unrolling the construction of $\hat{A}_t$ and looking into each multiplication of $\hat{K}$ matrices:

$$(\sqrt{\alpha} I + \beta_{t-1} q_{t-1} q_{t-1}^\top)(\sqrt{\alpha} I + \beta_t q_t q_t^\top) \tag{12}$$
$$= \alpha I + \left(\sqrt{\alpha} \beta_t q_t + \beta_{t-1} \beta_t (q_{t-1}^\top q_t) q_{t-1}\right) q_t^\top + \sqrt{\alpha} \beta_{t-1} q_{t-1} q_{t-1}^\top. \tag{13}$$

It is obvious that the rank of the sum of the last two terms is at most two. By repeating the multiplication $\tau$ times, we have $A_t - \alpha^{\tau/2} I$ at most of rank $\tau$, implying that $G_t - \alpha^\tau I = A_t A_t^\top - \alpha^\tau I$ has at most rank $2\tau$.

We argue that this practice does not capture the most useful information in $G_t$, as the optimal low-rank approximation $Q_t Q_t^\top$ is given by the spectral decomposition or singular value decomposition. We thus propose a QNG variant to maintain the significant low-rank approximation in an online fashion while keeping the space and time complexities linear in the number of model parameters.

## 2.3 Our approach: Ginger

Motivated by the above observation, we propose a novel QNG variant, called GINGER, that directly models a damped GGN matrix in the form of

$$G_{t,\gamma} = \gamma I + U_t \operatorname{diag}(\sigma_t) U_t^\top, \tag{14}$$

where $t$ is the update step and $\gamma$ is the damping strength. The second term $U_t \operatorname{diag}(\sigma_t) U_t^\top$ is a low-rank approximation of the GGN matrix (7), where $U_t \in \mathbb{R}^{d \times \tau}$ is a semi-orthogonal matrix,[1] and $0 \preceq \sigma_t \in \mathbb{R}^\tau$ is the vector of eigenvalues sorted in the descending order, for a rank of $\tau$.

Our approach generalizes He et al. (2022)'s QNG form as we also decompose the GGN matrix in a diagonal plus low-rank form. However, we directly model the low-rank part by spectral decomposition, whereas the diagonal is controlled by a damping hyperparameter $\gamma$. In this way, we can model the optimal approximation of the low-rank approximation by the Eckart–Young theorem Eckart & Young (1936).

**Querying the update direction.** Assuming the matrix $G_{t,\gamma}$ is already known, we are interested in the update direction $G_{t,\gamma}^{-1} g$ for any vector $g \in \mathbb{R}^d$, such as $g$ being the gradient of loss wrt the parameters. This can be obtained through the Woodbury matrix identity (Lemma 3):

$$
\begin{aligned}
G_{t,\gamma}^{-1} g &= (\gamma I + U_t \operatorname{diag}(\sigma_t) U_t^\top)^{-1} g \\
&= (\gamma^{-1} I - U_t \underbrace{(\gamma^2 I + \gamma \operatorname{diag}(\sigma_t))^{-1} \operatorname{diag}(\sigma_t)}_{K_{t,\gamma}} U_t^\top) g,
\end{aligned}
\tag{15}
$$

where $K_{t,\gamma}$ is a diagonal matrix that can be computed in $O(\tau)$ time (recall $\sigma_t \in \mathbb{R}^\tau$). Specifically, we have the following relationship

$$K_{t,\gamma}^{(i,i)} = \frac{\sigma_t^{(i)}}{\gamma^2 + \gamma \sigma_t^{(i)}} \tag{16}$$

between the $i$th elements in $K_{t,\gamma}$ and $\sigma_t$.

By computing the result from the right to the left, we can obtain the update direction in $O(d\tau)$ time.

**Update rules.** Assuming $G_{t-1,\gamma}$ is already constructed and the new gradient is $d_t$, we would like to use the moving average to update the undamped GGN approximation, i.e., the second term of Equation (14). Without restricting its rank (indicated by a tilde), we have

$$
\begin{aligned}
\tilde{G}_{t,\gamma}^{-1} &= (\gamma I + \alpha G_{t-1,0} + (1-\alpha) d_t d_t^\top)^{-1} \tag{EMA} \\
&= (\alpha G_{t-1,\gamma/\alpha} + (1-\alpha) d_t d_t^\top)^{-1} \\
&= \alpha^{-1} G_{t-1,\gamma/\alpha}^{-1} - \beta_t h_t h_t^\top \\
&= \gamma^{-1} I - (U_{t-1} (\alpha^{-1} K_{t-1,\gamma/\alpha}) U_{t-1}^\top + \beta_t h_t h_t^\top),
\end{aligned}
\tag{17}
$$

where $h_t := G_{t,\gamma/\alpha}^{-1} d_t$ and $\beta_t := \frac{\alpha^{-1}(1-\alpha)}{\alpha + (1-\alpha) h_t^\top d_t}$ are obtained by applying the Woodbury matrix identity again.

To maintain a low-rank approximation with rank $\tau$ mimicking the behavior of $\tilde{G}_{t,\gamma}^{-1}$, we would like to find a matrix $U_t \operatorname{diag}(\sigma_t) U_t^\top$ such that the error

$$
\begin{aligned}
\epsilon(U_t, \sigma_t) &:= \|\tilde{G}_{t,\gamma}^{-1} - (\gamma I + U_t \operatorname{diag}(\sigma_t) U_t^\top)^{-1}\|_2 \\
&= \|U_{t-1}(\alpha^{-1} K_{t-1,\gamma/\alpha}) U_{t-1}^\top + \beta_t h_t h_t^\top - U_t K_{t,\gamma} U_t^\top\|_2
\end{aligned}
$$

is minimized.

---

[1] Meaning $U_t^\top U_t = I$.

We observe that $U_t K_{t,\gamma} U_t^\top$ has a rank of at most $\tau$, so the optimal solution is given by the truncated SVD of $U_{t-1}(\alpha^{-1}K_{t-1,\gamma/\alpha})U_{t-1}^\top + \beta_t h_t h_t^\top$. The efficient computation of the SVD is deferred to the next subsection.

After obtaining $K_t$, we will find a new $\sigma_t$ for our GGN approximation in Equation (14). This can be done by matching the diagonal of $K_t$ with Equation (16), which yields

$$\sigma_t^{(i)} = \frac{\gamma^2 K_{t,\gamma}^{(i,i)}}{1 - \gamma K_{t,\gamma}^{(i,i)}} \tag{18}$$

for any $i \in \{1, \ldots, \tau\}$.

Note that $\sigma_t$ is guaranteed to be non-negative. This can be shown through Equation (17) by noticing that $\tilde{G}_{t,\gamma}$ is positive definite due to EMA, which implies:

$$0 \preceq U_{t-1}(\alpha^{-1}K_{t-1,\gamma/\alpha})U_{t-1}^\top + \beta_t h_t h_t^\top \prec \gamma^{-1}I \tag{19}$$

for any iteration step $t$. Here, matrix comparisons $A \preceq B$ and $A \prec B$ mean that $B - A$ is positive semi-definite and positive definite, respectively.

**Observation 2.** *By construction, Ginger finds the matrix $U_t K_{t,\gamma} U_t^\top$ that is the closest rank-$\tau$ at approximation to the target inverse term $M_t$ at every step. This means for any other rank-$\tau$ matrix $Z$, we have*

$$\|M_t - U_t K_{t,\gamma} U_t^\top\|_2 \leq \|M_t - Z\|_2.$$

*By contrast, QNG is not guaranteed to minimize the spectral norm distance.*

**Efficient SVD.** We now turn to the efficient computation of the SVD of $U_{t-1}(\alpha^{-1}K_{t-1,\gamma/\alpha})U_{t-1}^\top + \beta_t h_t h_t^\top$. By Equation (16), we know that $\alpha^{-1}K_{t-1,\gamma/\alpha}$ is a sorted diagonal matrix and $U_{t-1}$ is semi-orthogonal; therefore, the first term itself is in the SVD form. Observing $\beta_t h_t h_t^\top$ is rank-1, we can efficiently compute the new SVD. Specifically, we use the approach in Brand (2006), but our calculation will be simplified, as our SVD is essentially eigendecomposition because the matrix is positive semi-definite.

We first rewrite the update in the compact form:

$$U_{t-1}(\alpha^{-1}K_{t-1,\gamma/\alpha})U_{t-1}^\top + \beta_t h_t h_t^\top = \begin{pmatrix} U_{t-1} & h_t \end{pmatrix} \begin{pmatrix} \alpha^{-1}K_{t-1,\gamma/\alpha} & 0 \\ 0 & \beta_t \end{pmatrix} \begin{pmatrix} U_{t-1} & h_t \end{pmatrix}^\top.$$

We notice that $\begin{pmatrix} U_{t-1} & h_t \end{pmatrix}$ can be factorized as

$$\begin{pmatrix} U_{t-1} & p_t \end{pmatrix} \begin{pmatrix} I & U_{t-1}^\top h_t \\ 0 & r_t \end{pmatrix},$$

where $r_t = \|h_t - U_{t-1}U_{t-1}^\top h_t\|$ and $p_t = (h_t - U_{t-1}U_{t-1}^\top h_t)/r_t$. In this way, $\begin{pmatrix} U_{t-1} & p_t \end{pmatrix}$ is a semi-orthogonal matrix.

Therefore, we have new factorization

$$\begin{pmatrix} U_{t-1} & p_t \end{pmatrix} C_t \begin{pmatrix} U_{t-1} & p_t \end{pmatrix}^\top,$$

where $C_t$ is defined as

$$\begin{pmatrix} I & U_{t-1}^\top h_t \\ 0 & r_t \end{pmatrix} \begin{pmatrix} \alpha^{-1}K_{t-1,\gamma/\alpha} & 0 \\ 0 & \beta_t \end{pmatrix} \begin{pmatrix} I & U_{t-1}^\top h_t \\ 0 & r_t \end{pmatrix}^\top$$

with a shape of $(\tau + 1) \times (\tau + 1)$.

With $O(\tau^3)$ time, we can obtain the SVD of $C_t = VK'V^\top$. It is easy to see that

$$U_t'K'U_t'^\top = U_{t-1}(\alpha^{-1}K_{t-1,\gamma/\alpha})U_{t-1}^\top + \beta h_t h_t^\top,$$

---

**Algorithm 1:** Our approach: GINGER

---

**Input:** Decay rate $\alpha$, damping factor $\gamma$, rank $\tau$
**Input:** Initial parameters $\theta_0$
**def** `r1u`$(U, K, d)$:
    /* returns the SVD of $UKU^\top + dd^\top$ */
    $U, K \leftarrow \text{SVD}(UKU^\top + dd^\top)$               ▷ fast version
    **return** $U^{(:,1:\tau)}, K^{(1:\tau)}$               ▷ $O(\tau^2(d+\tau))$

**def** `drt`$(U, \sigma, g)$:
    /* calculates the update direction $(\gamma I + U \operatorname{diag}(\sigma) U^\top)^{-1} g$ */
    $\Sigma \leftarrow \operatorname{diag}(\sigma)$           ▷ $O(\tau)$
    $K_\gamma \leftarrow (\gamma^2 I + \gamma \Sigma)^{-1} \Sigma$           ▷ $O(\tau)$
    $g' \leftarrow U K_\gamma U^\top g$           ▷ $O(d\tau)$
    **return** $\gamma^{-1} g - g'$           ▷ $O(d)$

**def** `upd`$(U, \sigma, d)$:
    /* updates $U$ and $\sigma$ with $d$ */
    $h \leftarrow$ `drt`$(U, \alpha\sigma, d)$           ▷ $O(d\tau)$
    $\Sigma \leftarrow \operatorname{diag}(\sigma)$           ▷ $O(\tau)$
    $K_{\gamma/\alpha} \leftarrow ((\gamma/\alpha)^2 I + (\gamma/\alpha)\Sigma)^{-1} \Sigma$           ▷ $O(\tau)$
    $\beta \leftarrow \frac{\alpha^{-1}(1-\alpha)}{\alpha + (1-\alpha)h^\top d}$           ▷ $O(d)$
    $U, K \leftarrow$ `r1u`$(U, K_{\gamma/\alpha}, \sqrt{\beta}h)$           ▷ $O(\tau^2(d+\tau))$
    $\sigma \leftarrow \gamma^2 K (1 - \gamma K)^\dagger$           ▷ $O(\tau)$
    **return** $U, \sigma$

/* Initialization */
$U_0 \leftarrow$ random semi-orthogonal matrix
$\sigma_0 \leftarrow \mathbf{0}$
**for** $t \leftarrow 0 \ldots T-1$ **do**
    /* Update the optimizer state first */
    Learning rate schedule $\eta_t$
    $d_t \leftarrow$ Equation (8)
    $U_{t+1}, \sigma_{t+1} \leftarrow$ `upd`$(U_t, \sigma, d_t)$           ▷ $O(\tau^2(d+\tau))$
    /* Update parameters */
    $g_t \leftarrow$ `drt`$(U_{t+1}, \sigma_{t+1}, \nabla\mathcal{L}(\theta_t)), (U_{t+1}, \sigma_{t+1})$
    $\theta_{t+1} \leftarrow \theta_t - \eta_t g_t$
    $t \leftarrow t+1$
**return** $\theta_T$

---

where $U'_t = \begin{pmatrix} U_{t-1} & p_t \end{pmatrix} V$ is semi-orthogonal and $K'$ is a diagonal matrix. We thus conclude that $(U'_t, K')$ is the SVD of $U_{t-1}(\alpha^{-1}K_{t-1,\gamma/\alpha})U_{t-1}^\top + \beta_t h_t h_t^\top$.

Note that $U'_t \in \mathbb{R}^{d \times (\tau+1)}$ and $K'_t \in \mathbb{R}^{\tau+1}$ now have one more dimension than the previous iteration. We hence truncate the last column of $U'_t$ and the last element of $K'_t$ to obtain $U_t \in \mathbb{R}^{d \times \tau}$ and $K_t \in \mathbb{R}^\tau$, respectively. The truncated $U_t$ is still semi-orthogonal and $K_t$ is still diagonal. Finally, we use Equation (18) to translate $K_t$ back to $\sigma_t$.

The total computation of the process takes $O(d\tau^2 + \tau^3)$ time and $O(d\tau + \tau^2)$ space. Under a typical choice of $\tau$ where $\tau \ll \sqrt{d}$, we can simplify the complexities as $O(d\tau^2)$ and $O(d\tau)$ for time and space, respectively. Therefore, we conclude that the algorithm is linear to the number of parameters $d$ when $\tau \ll \sqrt{d}$.

The overall algorithm is summarized in Algorithm 1.

## 3 Theoretical analyses

We include the convergence proof of our method to show the sanity of our method. Before we present the theoretical analyses, we first show the following lemma that bounds the eigenvalues of the preconditioning matrix $G_{t,\gamma}$.

We make the following assumptions to obtain the convergence guarantee, where the first three are standard in the stochastic optimization analysis Bottou et al. (2018). The last assumption is especially for our method, which essentially assumes the gradient is finite.

**Assumption 1** (Lipschitz gradient)**.** The gradient of the loss function $\nabla_\theta \mathcal{L}$ is $L$-Lipschitz continuous with $L > 0$. This means that we have $\|\nabla\mathcal{L}(\theta_1) - \nabla\mathcal{L}(\theta_2)\| \leq L\|\theta_1 - \theta_2\|$ for all $\theta_1, \theta_2 \in \mathbb{R}^d$.

**Assumption 2** (Bounded gradient variance)**.** There exists a constant $M_g > 0$ such that for all $t \geq 1$,

$$\mathbb{E}_{(x_i,y_i)}[\|\nabla_\theta\mathcal{L}(\theta_t; x_i, y_i)\|^2] \leq M_g^2. \tag{20}$$

**Assumption 3** (Learning rate schedule)**.** The learning rate schedule satisfies the following conditions:

$$\sum_{t=1}^\infty \eta_t = \infty \quad \text{and} \quad \sum_{t=1}^\infty \eta_t^2 < \infty. \tag{21}$$

**Assumption 4** (Bounded Fisher vector variance)**.** The vector $d_t$ used for the rank-one Fisher update has a bounded second moment. There exists a constant $M_d > 0$ such that for all $t \geq 1$:

$$\mathbb{E}_t[\|d_t\|^2] \leq M_d^2.$$

Under these assumptions, we obtain the following lemmas:

**Lemma 1** (Eigenvalue bounds)**.** *For all $t \geq 0$, the eigenvalues of the inverse preconditioner are bounded. Specifically, $\lambda_{\max}(G_{t,\gamma}^{-1}) = \gamma^{-1}$ and there exists a constant $\lambda'_{\min} > 0$ such that $\mathbb{E}[\lambda_{\min}(G_{t,\gamma}^{-1})] \geq \lambda'_{\min}$.*

*Proof.* See Appendix B. □

**Lemma 2** (One-step progress)**.** *We have*

$$\mathbb{E}[\mathcal{L}(\theta_{t+1})] \leq \mathbb{E}[\mathcal{L}(\theta_t)] - \eta_t\,\mathbb{E}[\nabla\mathcal{L}(\theta_t)^\top G_{t+1,\gamma}^{-1}\nabla\mathcal{L}(\theta_t)] + \frac{L\eta_t^2 M_g^2}{2\gamma^2}.$$

*for all $t \geq 1$.*

*Proof.* See Appendix B. □

**Theorem 1.** *Under the above assumptions, we have*

$$\lim_{T\to\infty} \inf_{t=0,\ldots,T} \|\nabla\mathcal{L}(\theta_t)\|^2 = 0. \tag{22}$$

*Proof.* From Lemma 2, summing from $t = 0$ to $T - 1$ gives:

$$\sum_{t=0}^{T-1} \eta_t\,\mathbb{E}[\nabla\mathcal{L}(\theta_t)^\top G_{t+1,\gamma}^{-1}\nabla\mathcal{L}(\theta_t)] \leq \mathbb{E}[\mathcal{L}(\theta_0)] - \mathbb{E}[\mathcal{L}(\theta_T)] + \frac{LM_g^2}{2\gamma^2}\sum_{t=0}^{T-1}\eta_t^2.$$

Since $\mathcal{L}$ is bounded below by $\mathcal{L}_{\min}$ (Assumption 1) and $\sum \eta_t^2 < \infty$ (Assumption 3), the right-hand side is bounded above by a finite constant $C$ as $T \to \infty$.

$$\sum_{t=0}^\infty \eta_t\,\mathbb{E}[\nabla\mathcal{L}(\theta_t)^\top G_{t+1,\gamma}^{-1}\nabla\mathcal{L}(\theta_t)] \leq C < \infty.$$

Table 1: The results on the CIFAR-100 dataset.

| Optimizers | ResNet-18 | | | | | ResNet-50 | | | | |
|---|---|---|---|---|---|---|---|---|---|---|
| | Acc ↑ | $\mathcal{L}^{\text{val}}$ | $\hat{\mathcal{L}}^{\text{train}}_{10^{-6}}$ | FLOPs | Mem | Acc ↑ | $\mathcal{L}^{\text{val}}$ | $\hat{\mathcal{L}}^{\text{train}}_{10^{-6}}$ | FLOPs | Mem |
| Momentum | 71.89 | 1.320 | 10.47 | 2.21e9 | 3.15 | 74.42 | 1.276 | 11.47 | 1.13e10 | 5.92 |
| Adam | 71.61 | 1.249 | 9.57 | 2.36e9 | 3.16 | 71.59 | 1.241 | 60.65 | 1.16e10 | 6.50 |
| DNG | 71.04 | 1.433 | 12.82 | 3.02e9 | 3.19 | 70.72 | 1.394 | 61.01 | 1.43e10 | 6.55 |
| QNG ($\tau$=1) | 71.74 | 1.374 | 23.83 | 3.56e9 | 3.20 | 74.30 | 1.290 | 32.29 | 1.80e10 | 6.56 |
| QNG ($\tau$=2) | 72.30 | 1.347 | 23.97 | 3.70e9 | 3.25 | 73.92 | 1.255 | 42.12 | 1.83e10 | 6.78 |
| QNG ($\tau$=4) | 72.32 | 1.323 | 27.02 | 3.97e9 | 3.41 | 74.51 | 1.287 | 12.53 | 1.89e10 | 7.30 |
| QNG ($\tau$=8) | 71.98 | 1.318 | 15.91 | 4.51e9 | 3.59 | 75.12 | 1.311 | 6.98 | 2.00e10 | 7.66 |
| QNG ($\tau$=16) | 71.76 | 1.290 | 13.60 | 5.59e9 | 3.92 | 74.53 | 1.254 | 6.35 | 2.23e10 | 8.65 |
| QNG ($\tau$=32) | 71.95 | 1.321 | 18.08 | 7.74e9 | 4.60 | 74.27 | 1.267 | 9.37 | 2.68e10 | 10.66 |
| GINGER ($\tau$=1) | 72.04 | 1.265 | 117.8 | 3.77e9 | 3.19 | 74.54 | 1.269 | 37.07 | 1.84e10 | 6.60 |
| GINGER ($\tau$=2) | 72.66 | 1.270 | 22.78 | 3.80e9 | 3.25 | 74.99 | 1.245 | 11.86 | 1.85e10 | 6.66 |
| GINGER ($\tau$=4) | 72.90 | 1.263 | 13.37 | 4.08e9 | 3.33 | 75.83 | 1.249 | 3.90 | 1.91e10 | 6.91 |
| GINGER ($\tau$=8) | 72.89 | 1.242 | 11.18 | 4.66e9 | 3.56 | 75.73 | 1.255 | 4.26 | 2.03e10 | 7.36 |
| GINGER ($\tau$=16) | 73.17 | 1.236 | **6.41** | 5.82e9 | 3.90 | 75.53 | **1.202** | **3.82** | 2.28e10 | 8.20 |
| GINGER ($\tau$=32) | 73.15 | **1.210** | 9.88 | 8.16e9 | 4.61 | 75.48 | 1.216 | 5.27 | 2.77e10 | 10.17 |

We use a probabilistic splitting. Define the "good" event $\mathcal{E}_t := \{\lambda_{\max}(\sigma_{t+1}) \leq \kappa\}$, where we set $\kappa = 2M_d^2$, corresponding to the choice $\epsilon = 1/2$ in the proof of Lemma 1. This ensures $\Pr(\mathcal{E}_t) \geq 1/2$. When $\mathcal{E}_t$ occurs, $\lambda_{\min}(G^{-1}_{t+1,\gamma}) = \frac{1}{\gamma + \lambda_{\max}(\sigma_{t+1})} \geq \frac{1}{\gamma + \kappa}$.

We can split the expectation:

$$
\begin{aligned}
&\mathbb{E}[\nabla\mathcal{L}(\theta_t)^\top G^{-1}_{t+1,\gamma}\nabla\mathcal{L}(\theta_t)] \\
=&\mathbb{E}[\nabla\mathcal{L}(\theta_t)^\top G^{-1}_{t+1,\gamma}\nabla\mathcal{L}(\theta_t) \mid \mathcal{E}_t]\Pr(\mathcal{E}_t) + \mathbb{E}[\nabla\mathcal{L}(\theta_t)^\top G^{-1}_{t+1,\gamma}\nabla\mathcal{L}(\theta_t) \mid \mathcal{E}_t^c]\Pr(\mathcal{E}_t^c) \\
\geq&\mathbb{E}[\nabla\mathcal{L}(\theta_t)^\top G^{-1}_{t+1,\gamma}\nabla\mathcal{L}(\theta_t) \mid \mathcal{E}_t]\Pr(\mathcal{E}_t) && \text{(Since the term is non-negative)} \\
\geq&\mathbb{E}[\lambda_{\min}(G^{-1}_{t+1,\gamma})\|\nabla\mathcal{L}(\theta_t)\|^2 \mid \mathcal{E}_t]\Pr(\mathcal{E}_t) \\
\geq&\frac{1}{\gamma + \kappa}\mathbb{E}[\|\nabla\mathcal{L}(\theta_t)\|^2 \mid \mathcal{E}_t]\Pr(\mathcal{E}_t).
\end{aligned}
$$

The term $\nabla\mathcal{L}(\theta_t)$ is a function of the history up to step $t-1$. The event $\mathcal{E}_t$ depends on the random sample $d_t$, which is drawn independently of that history. Consequently, the random variable $\|\nabla\mathcal{L}(\theta_t)\|^2$ is independent of the event $\mathcal{E}_t$. For a random variable $X$ that is independent of an event $A$, we have $\mathbb{E}[X \mid A] = \mathbb{E}[X]$. Therefore:

$$\mathbb{E}[\|\nabla\mathcal{L}(\theta_t)\|^2 \mid \mathcal{E}_t] = \mathbb{E}[\|\nabla\mathcal{L}(\theta_t)\|^2].$$

This gives us:

$$\mathbb{E}[\nabla\mathcal{L}(\theta_t)^\top G^{-1}_{t+1,\gamma}\nabla\mathcal{L}(\theta_t)] \geq \frac{1}{\gamma + \kappa}\mathbb{E}[\|\nabla\mathcal{L}(\theta_t)\|^2] \cdot \frac{1}{2}.$$

Let $\lambda'_{\min} = \frac{1}{2(\gamma+\kappa)} = \frac{1}{2(\gamma+2M_d^2)} > 0$. Substituting this back into the sum:

$$\sum_{t=0}^{\infty} \eta_t \lambda'_{\min} \mathbb{E}[\|\nabla\mathcal{L}(\theta_t)\|^2] \leq C < \infty.$$

Since $\lambda'_{\min} > 0$ is a constant, we have shown that $\sum_{t=0}^{\infty} \eta_t \mathbb{E}[\|\nabla\mathcal{L}(\theta_t)\|^2] < \infty$. Given that $\mathbb{E}[\|\nabla\mathcal{L}(\theta_t)\|^2] \geq 0$ and $\sum_{t=0}^{\infty} \eta_t = \infty$ (Assumption 3), a standard result from analysis states that for the sum to be finite, it must be that $\liminf_{t\to\infty} \mathbb{E}[\|\nabla\mathcal{L}(\theta_t)\|^2] = 0$. $\square$

We provide the theoretical analysis here to show the sanity of our approximation. It is worth noting that the convergence result holds regardless of the convexity of the objective.

Table 2: The results on the XSUM dataset.

| Optimizers | LoRA | | | | Full | | | |
|---|---|---|---|---|---|---|---|---|
| | R1/R2/RL ↑ | $\hat{\mathcal{P}}^{\text{train}}$ | FLOPs | Mem | R1/R2/RL ↑ | $\hat{\mathcal{P}}^{\text{train}}$ | FLOPs | Mem |
| Adam | 31.06/9.09/24.10 | 10.236 | 5.61e9 | 2.67 | 32.61/10.40/25.48 | 3.747 | 5.42e9 | 2.06 |
| Momentum | 27.39/6.64/20.81 | 12.566 | 5.61e9 | 2.67 | 29.90/8.28/23.07 | 4.831 | 4.60e9 | 1.98 |
| QNG ($\tau$=1) | 29.01/7.63/22.32 | 11.775 | 5.62e9 | 2.67 | 29.96/8.34/23.11 | 4.938 | 8.60e9 | 3.45 |
| QNG ($\tau$=2) | 28.99/7.64/22.28 | 11.658 | 5.63e9 | 2.68 | 29.94/8.39/23.17 | 4.778 | 9.05e9 | 3.75 |
| QNG ($\tau$=4) | 28.95/7.74/22.42 | 11.646 | 5.66e9 | 2.69 | 29.98/8.30/23.13 | 4.826 | 1.22e10 | 4.66 |
| QNG ($\tau$=8) | 29.01/7.74/22.42 | 11.822 | 5.74e9 | 2.71 | 29.84/8.29/23.13 | 4.860 | 2.21e10 | 5.89 |
| QNG ($\tau$=16) | 28.98/7.67/22.29 | 11.717 | 6.05e9 | 2.74 | 29.85/8.37/23.13 | 4.870 | 5.06e10 | 9.31 |
| QNG ($\tau$=32) | 29.00/7.66/22.28 | 11.752 | 7.15e9 | 2.77 | 29.91/8.34/23.05 | 4.879 | 9.76e10 | 16.51 |
| GINGER ($\tau$=1) | 31.03/9.03/24.03 | 10.278 | 5.63e9 | 2.67 | 32.73/10.51/25.64 | 4.212 | 1.00e10 | 3.43 |
| GINGER ($\tau$=2) | 31.14/9.07/24.11 | 10.298 | 5.64e9 | 2.68 | 32.93/10.68/25.85 | 3.532 | 1.10e10 | 3.67 |
| GINGER ($\tau$=4) | 31.07/9.10/24.10 | 10.268 | 5.68e9 | 2.68 | 32.85/10.71/25.85 | 3.618 | 1.43e10 | 4.35 |
| GINGER ($\tau$=8) | 31.07/9.10/24.10 | 10.196 | 5.77e9 | 2.70 | 32.81/10.60/25.72 | 3.564 | 2.41e10 | 6.48 |
| GINGER ($\tau$=16) | 30.93/8.96/24.04 | 10.176 | 6.08e9 | 2.74 | 32.88/10.60/25.78 | 3.370 | 5.51e10 | 9.77 |
| GINGER ($\tau$=32) | 31.03/9.10/24.08 | 10.044 | 7.18e9 | 2.77 | 32.82/10.66/25.75 | 3.418 | 16.33e10 | 18.04 |

## 4 Experiments

In this section, we conduct experiments on different tasks and model architectures to verify the effectiveness of GINGER. For task selection, we consider image classification and conditional language modeling, which are two symbolic benchmarks in deep learning. For the baselines, we only consider the methods that are able to achieve linear memory and time complexity, which include the first-order methods and the quasi-second-order methods. For the former, we consider the standard momentum method and the well-established Adam optimizer Kingma & Ba (2015). For the latter, we mainly consider the quasi-natural gradient (QNG) method recently proposed by He et al. (2022).

### 4.1 Image classification

**Dataset.** We use the CIFAR-100 (Krizhevsky et al., 2009) image classification dataset, which contains 50K training and 10K test images. Each data point is a $32 \times 32$ RGB image belonging to one of the 100 classes.

**Models.** We consider popular convolutional neural network (CNN) architectures for image classification, namely, ResNet-18 (11M parameters) and ResNet-50 (24M parameters) (He et al., 2016).

**Training details.** We use the standard data augmentation and normalization for training He et al. (2016). We set the coefficients of first moment and second moment as 0.9 and 0.99, respectively, for all optimizers. We tune the learning rate in the set of $\{1, 5\} \times 10^{\{-1, -2, -3, -4\}}$ for all optimizers on a subset of the validation set. Additionally, we test a variant of the quasi-second-order method that maintains the diagonal elements of the FIM. It is labeled as DNG in the experiments. After tuning, we fix the learning rate for each optimizer across all variations of it. To rule out the effect of learning rate scheduling and weight decay, we do not use them in our experiments. We train all models for 200 epochs with a batch size of 128.

**Evaluation metrics.** We report the best validation accuracy and the corresponding evaluating loss $\mathcal{L}^{\text{val}}$. To get more insights into the training process, we also report the minimum training loss (scaled by $10^{-6}$ for readability) on sampled batches. In addition, we calculate floating point operations per iteration (FLOPs) and the peak memory usage with the JAX profiling tool (Bradbury et al., 2018).

**Results.** The main results are summarized in Table 1. We can see that GINGER achieves the best validation accuracy on both ResNet-18 and ResNet-50. In addition, GINGER achieves the best training loss on ResNet-18 and the second-best training loss on ResNet-50. These results indicate that GINGER is able to achieve better generalization performance than other optimizers. In terms of FLOPs and memory usage, GINGER inevitably

requires more FLOPs and memory than the first-order methods like Momentum or Adam. However, it is still able to achieve linear memory and time complexity, which is much more efficient than the quasi-second-order methods.

An interesting observation is that when $\tau$ grows larger, the performance of GINGER generally increases. This is because a larger $\tau$ leads to a more accurate approximation of the preconditioning matrix. However, the performance saturates when $\tau$ is large enough, as the generalized Gauss–Newton matrix heavily depends on the leading eigenvalues, corroborating findings in prior work (Feinberg et al., 2023).

Although a larger $\tau$ generally yields better approximation, it also leads to more FLOPs (quadratic increase) and memory (linear increase). As mentioned in Section 2, however, we typically have $\tau \ll \sqrt{d}$, so our approach does not add to the complexity much compared with other parts of the learning algorithm, such as forward and backward propagation.

## 4.2 Conditional language modeling

Language modeling is another well-established task in deep learning, with the tremendous success of large language models like GPT-3 (Brown et al., 2020). In this paper, we specifically consider conditional language modeling, namely, text summarization, as it is easier to evaluate.

**Dataset.** We use the XSUM dataset (Narayan et al., 2018) in our experiments. XSUM is a summarization dataset that contains 204K training samples, 11K validation samples, and 11K test samples. Each sample is a news article with a summary. The task is to generate a summary of the article.

**Models.** We use the standard Transformer as our model architecture. Specifically, we load a pre-trained T5-small model (Raffel et al., 2020) and fine-tune it with the XSUM dataset. The model has around 60M parameters in total.

In addition to the standard full-parameter fine-tuning, we also consider the setting where only low-rank adapters (LoRA) (Hu et al., 2022) are fine-tuned. It has gained increasing attention recently because it is able to achieve comparable performance with much fewer parameters, making it an ideal choice for fine-tuning large language models.

**Training details.** For each sample, we first tokenize the source and target sentences with the T5 tokenizer. We then truncate the source to 512 tokens and the target to 128 tokens.

Most of the hyper-parameters are tuned in the same way as in the image classification task. In addition, we set the rank of each attention adapter as 8 for the LoRA setting. We train all models for 1 epoch with a batch size of 4.

**Evaluation metrics.** We report the best rouges scores (Lin, 2004), including ROUGE-1, ROUGE-2, and ROUGE-L individually. These scores represent the overlap between the generated summary and the ground-truth summary. They are widely used in summarization tasks. We also report the training perplexity $\hat{\mathcal{P}}^{\text{train}}$ and the evaluating loss $\mathcal{L}^{\text{val}}$. Similar to image classification, we also report FLOPs and the peak memory usage for each optimizer.

**Results.** For the full-parameter fine-tuning setting, there is a clear trend that GINGER achieves better performance than other optimizers. Especially, the larger $\tau$ leads to lower training loss during the training process. Further, the lower loss translates to better rouges scores in general. This indicates that GINGER is able to maintain a reasonable generalization ability.

For the LoRA fine-tuning setting, GINGER also achieves a marginally better performance. However, the performance of GINGER is not as good as the full-parameter fine-tuning setting. We hypothesize that this is because the LoRA weights are generally easier to optimize with their low-rank structure, making the curvature information less important. Nevertheless, we argue in this case that GINGER is agnostic to architectural modifications or gradient calculation methods, making it a more general optimizer.

### 4.3 Analyses.

**Sensitive of $\tau$ and $\alpha$.** We conduct experiments to analyze the sensitivity of $\tau$ and $\alpha$ on the image classification task. We use ResNet-18 as the model architecture and CIFAR-100 as the dataset. All the other hyper-parameters are the same as in the main experiments. We report the best validation accuracy and the minimum training loss in Table 3 and Table 4, respectively.

<table>
<tr><td colspan="4">Table 3: Validation accuracy (%).</td><td colspan="4">Table 4: Training loss ($10^{-6}$).</td></tr>
<tr><td>$\alpha\backslash\tau$</td><td>2</td><td>4</td><td>8</td><td>$\alpha\backslash\tau$</td><td>2</td><td>4</td><td>8</td></tr>
<tr><td>0.9</td><td>73.24</td><td>73.35</td><td>73.15</td><td>0.9</td><td>14.82</td><td>6.03</td><td>10.62</td></tr>
<tr><td>0.99</td><td>72.66</td><td>72.90</td><td>72.89</td><td>0.99</td><td>22.78</td><td>13.37</td><td>11.18</td></tr>
<tr><td>0.999</td><td>72.73</td><td>73.14</td><td>72.57</td><td>0.999</td><td>52.98</td><td>11.92</td><td>17.55</td></tr>
</table>

As shown, the training losses vary only slightly (on the order of $10^{-5}$), suggesting that performance is not significantly degraded even under hyperparameter perturbations. This indicates that GINGER is robust to the choice of hyperparameters in the sense that it maintains stable training without divergence across a wide range of $\alpha$ and $\tau$ values.

## 5 Discussion

There has been a long history of approximating second-order optimization methods. The most popular ones are BFGS (Nocedal & Wright, 1999) and L-BFGS (Liu & Nocedal, 1989), which approximate the inverse of the Hessian matrix with reasonably large memory and time complexity. However, they are not suitable for non-convex functions with non-PSD Hessian matrices.

The generalized Gauss–Newton method as an approximation of the Hessian matrix is guaranteed to have the PSD preconditioning. However, materializing the exact Gauss–Newton method is not practical for modern deep learning. Martens et al. (2010), which uses the conjugate gradient method to solve the linear system with the Hessian-vector product. However, the memory and time complexity of the Hessian-free method are still too high for modern deep learning.

To further reduce the excessive memory and time complexity, Martens & Grosse (2015) proposed KFAC, which approximates the Fisher information matrix with Kronecker factors. However, it is restricted to certain model architectures. Moreover, even with the block-diagonal approximation or its variant Ba et al. (2017), the time complexity of KFAC takes $O(n^3)$ for each layer with a hidden size of $n$. This translates to at least $O(d^{1.5})$ for the total parameter size $d$ of the model, making it strictly superlinear in model size.

Recently, He et al. (2022) proposed a quasi-natural gradient method, which approximates the Fisher information matrix with linear complexity. As discussed in this paper, we show that the quasi-natural gradient method is equivalent to an identity matrix plus a low-rank matrix. However, the low-rank matrix might not capture the curvature information well, which leads to a non-informative preconditioner. In contrast, our method directly minimizes the norm difference between the inverses of the next EMA and approximation. As confirmed by experiments, our method is more effective than the quasi-natural gradient method.

Instead of the generalized Gauss–Newton matrix, it is also possible to use $(\sum_{i=1}^{t}[g_i g_i^\top])^{1/2}$ as $G_t$, known as AdaGrad (Duchi et al., 2011). The full-matrix AdaGrad requires quadratic memory and cubic time, not scalable to large models. To reduce the complexity, Gupta et al. (2018) proposed Shampoo, which approximates the full-matrix AdaGrad with Kronecker factors. However, the time complexity is $O(d^{1.5})$, which does not scale well to large models.

## 6 Conclusion

**Summary.** In this work, we propose GINGER, an efficient curvature approximation with linear complexity for general neural networks. Specifically, it is based on the eigendecomposition of the inverse generalized Gauss–Newton matrix. We show convergence of GINGER for non-convex objectives. Experiments on different tasks with different model architectures verify the effectiveness of our method.

**Future directions.** In this work, we build the convergence proof of GINGER to show the sanity of our method. However, we only show its benefits empirically and do not attempt to obtain the asymptotic convergence rates. This is because they typically require additional assumptions of the loss function. We leave this direction to future work. In addition, the time complexity is $O(d\tau^2)$ for $\tau \ll \sqrt{d}$, which may grow quadratically in $\tau$. We hope to reduce the complexity to $O(d\tau)$ to make it scalable to large models.

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

## A  Technical Lemmas

**Lemma 3** (Woodbury matrix identity). *For an invertible matrix $A \in \mathbb{R}^{n \times n}$ and matrices $U \in \mathbb{R}^{n \times k}$ and $V \in \mathbb{R}^{k \times n}$, where the matrix $(I_k + VA^{-1}U)$ is also invertible, the identity holds:*

$$(A + UV)^{-1} = A^{-1} - A^{-1}U(I_k + VA^{-1}U)^{-1}VA^{-1}.$$

*When the rank $k$ is much smaller than the dimension $n$ (i.e., $k \ll n$), this formula avoids the costly inversion of the large $n \times n$ matrix $A + UV$.*

## B  Proofs

In this section, we provide proofs for Lemma 1 and Lemma 2.

### B.1  Proof of Lemma 1

**Lemma 1** (Eigenvalue bounds). *For all $t \geq 0$, the eigenvalues of the inverse preconditioner are bounded. Specifically, $\lambda_{\max}(G_{t,\gamma}^{-1}) = \gamma^{-1}$ and there exists a constant $\lambda'_{\min} > 0$ such that $\mathbb{E}[\lambda_{\min}(G_{t,\gamma}^{-1})] \geq \lambda'_{\min}$.*

*Proof.* The upper bound follows from the construction: $G_{t,\gamma} = \gamma I + U_t \operatorname{diag}(\sigma_t) U_t^\top \succeq \gamma I$, which gives $\lambda_{\max}(G_{t,\gamma}^{-1}) = \gamma^{-1}$.

For the lower bound, we have $\lambda_{\min}(G_{t,\gamma}^{-1}) = \frac{1}{\lambda_{\max}(G_{t,\gamma})} = \frac{1}{\gamma + \lambda_{\max}(\sigma_t)}$. The eigenvalues in $\sigma_t$ are the top-$\tau$ eigenvalues of $G_{t,0}$, so $\lambda_{\max}(\sigma_t) \leq \lambda_{\max}(G_{t,0}) \leq \operatorname{Tr}(G_{t,0})$. From the EMA update and taking total expectations:

$$\mathbb{E}[\operatorname{Tr}(G_{t,0})] = \alpha \, \mathbb{E}[\operatorname{Tr}(G_{t-1,0})] + (1 - \alpha) \, \mathbb{E}[\|d_t\|^2] \leq \alpha \, \mathbb{E}[\operatorname{Tr}(G_{t-1,0})] + (1 - \alpha)M_d^2.$$

Since $\alpha \in (0, 1)$ and assuming $G_{0,0} = 0$, by induction we have $\mathbb{E}[\operatorname{Tr}(G_{t,0})] \leq M_d^2$ for all $t$. For any $\epsilon \in (0, 1)$, by Markov's inequality, we can choose $\kappa = M_d^2/\epsilon$ such that $\Pr[\lambda_{\max}(\sigma_t) \geq \kappa] \leq \Pr[\operatorname{Tr}(G_{t,0}) \geq \kappa] \leq \frac{\mathbb{E}[\operatorname{Tr}(G_{t,0})]}{\kappa} \leq \epsilon$. Now we can lower bound the expected minimum eigenvalue:

$$
\begin{aligned}
\mathbb{E}[\lambda_{\min}(G_{t,\gamma}^{-1})] &= \mathbb{E}\left[\frac{1}{\gamma + \lambda_{\max}(\sigma_t)}\right] \\
&\geq \mathbb{E}\left[\frac{1}{\gamma + \lambda_{\max}(\sigma_t)} \mathbf{1}_{\{\lambda_{\max}(\sigma_t) < \kappa\}}\right] \\
&\geq \frac{1}{\gamma + \kappa} \Pr[\lambda_{\max}(\sigma_t) < \kappa] \\
&\geq \frac{1}{\gamma + \kappa}(1 - \epsilon) = \frac{1}{\gamma + M_d^2/\epsilon}(1 - \epsilon).
\end{aligned}
$$

Since this holds for any $\epsilon \in (0, 1)$, we can choose $\epsilon = 1/2$ for concreteness, which yields a positive constant $\lambda'_{\min} = \frac{1}{2(\gamma + 2M_d^2)} > 0$. $\qquad\square$

### B.2  Proof of Lemma 2

**Lemma 2** (One-step progress). *We have*

$$\mathbb{E}[\mathcal{L}(\theta_{t+1})] \leq \mathbb{E}[\mathcal{L}(\theta_t)] - \eta_t \, \mathbb{E}[\nabla\mathcal{L}(\theta_t)^\top G_{t+1,\gamma}^{-1} \nabla\mathcal{L}(\theta_t)] + \frac{L\eta_t^2 M_g^2}{2\gamma^2}.$$

*for all $t \geq 1$.*

*Proof.* From the Lipschitz continuity of the gradient (Assumption 1):

$$\mathcal{L}(\theta_{t+1}) \leq \mathcal{L}(\theta_t) + \langle \nabla\mathcal{L}(\theta_t), \theta_{t+1} - \theta_t \rangle + \frac{L}{2}\|\theta_{t+1} - \theta_t\|^2.$$

Taking the total expectation and substituting the update rule $\theta_{t+1} - \theta_t = -\eta_t G_{t+1,\gamma}^{-1} g_t$:

$$\mathbb{E}[\mathcal{L}(\theta_{t+1})] \leq \mathbb{E}[\mathcal{L}(\theta_t)] - \eta_t \mathbb{E}[\langle \nabla\mathcal{L}(\theta_t), G_{t+1,\gamma}^{-1} g_t \rangle] + \frac{L\eta_t^2}{2}\mathbb{E}[\|G_{t+1,\gamma}^{-1} g_t\|^2].$$

For the cross-term, we use the tower property. We first condition on the history up to step $t$ ($\mathcal{F}_t$) and the random vector $d_t$, which makes $G_{t+1,\gamma}$ fixed.

$$\mathbb{E}[\langle \nabla\mathcal{L}(\theta_t), G_{t+1,\gamma}^{-1} g_t \rangle] = \mathbb{E}\left[\mathbb{E}[\langle \nabla\mathcal{L}(\theta_t), G_{t+1,\gamma}^{-1} g_t \rangle \mid \mathcal{F}_t, d_t]\right].$$

Inside the inner expectation, and by the independence of the mini-batches for $g_t$ and $d_t$:

$$\mathbb{E}[\langle \nabla\mathcal{L}(\theta_t), G_{t+1,\gamma}^{-1} g_t \rangle \mid \mathcal{F}_t, d_t] = \langle \nabla\mathcal{L}(\theta_t), G_{t+1,\gamma}^{-1}\mathbb{E}[g_t \mid \mathcal{F}_t, d_t] \rangle = \nabla\mathcal{L}(\theta_t)^\top G_{t+1,\gamma}^{-1}\nabla\mathcal{L}(\theta_t).$$

Taking the outer expectation gives $\mathbb{E}[\nabla\mathcal{L}(\theta_t)^\top G_{t+1,\gamma}^{-1}\nabla\mathcal{L}(\theta_t)]$. For the final term, using $\lambda_{\max}(G_{t+1,\gamma}^{-1}) = \gamma^{-1}$ and Assumption 2

$$\mathbb{E}[\|G_{t+1,\gamma}^{-1} g_t\|^2] \leq \mathbb{E}[\lambda_{\max}(G_{t+1,\gamma}^{-1})^2\|g_t\|^2] \leq \frac{1}{\gamma^2}\mathbb{E}[\|g_t\|^2] \leq \frac{M_g^2}{\gamma^2}.$$

The result follows by combining the terms. □

