# OpenReview forum: "Ginger: An Efficient Curvature Approximation with Linear Complexity for General Neural Networks"
_TMLR — Rejected by TMLR_

### Review · Reviewer_y4Hr · 2025-06-08

**Summary Of Contributions:**

This work introduces Ginger, an alternative approximation to the generalized Gauss-Newton matrix, which itself is an approximation to the Hessian of a loss function. The update is based on an identity plus low-rank approximation, where the low-rank component (inverse) can be updated using a rank-one addition plus a truncated SVD, which can be done with low complexity. The authors test the proposed method on two tasks and demonstrate better-than-Adam performance with around double the computing power required.

**Audience:**

Yes

**Broader Impact Concerns:**

-

**Claims And Evidence:**

No

**Requested Changes:**

**Critical**

* Address raised issues
* Add additional methods in empirical section for fair comparison

**Strengthening**

* Strengthen convergence result, e.g. converting residual convergence from liminf to inf

**Strengths And Weaknesses:**

**Strengths**

* This work places a heavy emphasis on the computational aspects of computing the GGN matrix, and the complexities of each step such as approximating Hessian-vector products are well displayed.
* The method is well-motivated and well-presented in the main text.

**Major**

* The convergence result is rather weak. Firstly, it has a liminf structure. Moreover, it follows a very similar structure of SGD and disregards the supposed approximation to the GGN matrix, and since the preconditioning is close to identity, the results all transfer with some constant changes. This would be more interesting if the convergence could be made stronger or linked with the approximation of the GGN matrix.
* The empirical component compares with only one other Gauss-Newton method. Given that there are other methods mentioned in the paper such as diagonal approximation, the authors should add this or address why such comparison is absent.

**Minor**

* (p.2) Please clarify the connection between GGN and natural gradient when your distribution $r$ is in the exponential family.
* Eq (13) seems to have a problem, the middle term is a vector/has an incorrect vector-vector product
* Please define semi-orthogonal of $U_t$ in Section 2.3, presumably $U_t^\top U_t = I_\tau$?
* It is probably worth putting the exact forms of the Woodbury matrix identity/Sherman-Morrison formula, perhaps adding the different forms in the appendix. The linear algebra is a bit unclear in its current form.
* Isn't Lemma 1 trivial? Since $G_{t,\gamma}= \gamma_I + U_t D U_t^\top$, the second term is a positive semidefinite matrix, so $G_{t,\gamma}\succeq \gamma I $ hence $G_{t,\gamma}^{-1}\preceq \gamma^{-1} I$.
* Table 1: in the caption "we use green and red", color seems to be missing.
* Section 5 discussion could be moved to after the introduction.
* Broader impact default text section is still present.

**Typos**

* (p.4) "approximation with rank $\tau$ mimic ..." -> "mimicking"

---

> ### Author Response · Authors · 2025-07-15
>
> We thank the reviewers for the detailed and constructive comments.
>
> > The convergence result is rather weak. Firstly, it has a liminf structure. Moreover, it follows a very similar structure of SGD and disregards the supposed approximation to the GGN matrix, and since the preconditioning is close to identity, the results all transfer with some constant changes. This would be more interesting if the convergence could be made stronger or linked with the approximation of the GGN matrix.
>
> We have substantially revised our theoretical analysis in response to this concern. The updated theory replaces the strong noise assumption with a more realistic variance-bounded assumption and establishes a more meaningful convergence result. The new analysis better connects the role of our preconditioner to the underlying GGN structure and highlights how it improves optimization beyond a simple SGD-like behavior.
>
> > The empirical component compares with only one other Gauss-Newton method. Given that there are other methods mentioned in the paper such as diagonal approximation, the authors should add this or address why such comparison is absent.
>
> Thank you for the suggestion. We have now included results for diagonal approximation in Table 1. Across both models, the diagonal method underperforms compared to our approach and the other baselines, supporting our claims about the advantages of Ginger.
>
> > (p.2) Please clarify the connection between GGN and natural gradient when your distribution r is in the exponential family.
>
> We have clarified this in the revision. When the output distribution $r$ belongs to the exponential family, the GGN matrix coincides with the Fisher Information Matrix, making GGN equivalent to natural gradient in this case.
>
> > Eq (13) seems to have a problem, the middle term is a vector/has an incorrect vector-vector product
>
> Thanks for pointing this out. We have corrected the typo in the revised version.
>
> > Please define semi-orthogonal of Ut in Section 2.3, presumably Ut⊤Ut=Iτ?
>
> Yes, that is correct. We now explicitly define this property in a footnote in Section 2.3.
>
> > It is probably worth putting the exact forms of the Woodbury matrix identity/Sherman-Morrison formula, perhaps adding the different forms in the appendix. The linear algebra is a bit unclear in its current form.
>
> Thanks for the suggestion. We have added a lemma in the Appendix with the explicit forms of the Woodbury identity and related matrix inversion results for clarity.
>
> > Isn't Lemma 1 trivial? Since $G_{t,\gamma}= \gamma I+U_t D U_t^\top$, the second term is a positive semidefinite matrix, so $G_{t,\gamma} \ge \gamma I$ hence $G_{t,\gamma}^{-1}\le \gamma^{−1}I$.
>
> We have revised the lemma under the new, weaker assumption to offer non-trivial insights under more realistic conditions.
>
> > Table 1: in the caption "we use green and red", color seems to be missing.
> > Section 5 discussion could be moved to after the introduction.
> > Broader impact default text section is still present.
> > Typo (p.4) "approximation with rank τ mimic ..." -> "mimicking"
>
> Thanks. We have fixed the writing errors in the paper.
>
>
> ---
>
> **Summary**
>
> The reviewers raised concerns about weak convergence theory and incomplete empirical comparisons, along with various technical clarifications. We have addressed these by substantially revising our theoretical analysis with stronger convergence results that better connect to reality, adding missing baseline comparisons including diagonal approximation, and providing all requested technical clarifications and corrections. We appreciate the thorough review and remain committed to further enhancing the work based on any additional feedback.

---

### Review · Reviewer_UjGc · 2025-06-19

**Summary Of Contributions:**

This paper aims to improve the practicability of Second-order optimization approaches in the context of  deep learning by approximating the preconditioning matrices. It proposes Ginger, an eigen-decomposition for the inverse of the generalized Gauss-Newton matrix. The proposed method has linear memory and time complexity for each iteration in theory, by maintaining the  inverse of the conditioning matrix directly.  It also provides the convergence result of Ginger for non-convex objectives. The experiments are conducted on image classification and conditional language modeling to  verify the effectiveness of the proposed method.

**Audience:**

Yes

**Claims And Evidence:**

Yes

**Requested Changes:**

In Section 4.3, this paper claims "GINGER is robust to the hyper-parameters". However, from the results shown in Table 4, the training loss is very sensitive to $\alpha$ and $\tao$.  I think this paper should explain it.

**Strengths And Weaknesses:**

Strength:

This paper provides good theoretical and technical contribution in analyzing the proposed method. The motivation is clear and the proposed method is intuitively provides better approximation to the preconditioning matrix, comparing to the method in He et al. (2022). The presentation is very clear, and this paper well organizes the background to understand the details of this paper.



Weaknesses:
1. This paper claims that "we discuss in Observation 1 that this approximation tends to be inaccurate, leading to a worse approximation". It is better for this paper to provide more empirical results to support this claim. Even thought this paper provides high-level thoughts in illustrating the approximation relating to the rank of a matrix, it is not clear whether the approximation is more/less accurate in practice (no quantitatively indicators provided) . Another questions are  whether more accurate approximation has better performances (e.g., better optimization performance, or better generalization performance). It is better for this paper to provide these results.



2. In Section 4.3, this paper claims "GINGER is robust to the hyper-parameters". However, from the results shown in Table 4, the training loss is very sensitive to $\alpha$ and $\tao$.  I think this paper should explain it.



3. It is not clear whether the proposed method is effective than the block-wise FIM approximation(e.g., [1])?



Other minors:

(1) "Putting (6) into (5)", should be  "Putting (5) into (4)"



Ref:

[1] Distributed second-order optimization using kronecker-factored approximations, ICLR 2017.

---

> ### Author Response · Authors · 2025-07-15
>
> We thank the reviewers for their valuable feedback and constructive suggestions.
>
> > This paper claims that "we discuss in Observation 1 that this approximation tends to be inaccurate, leading to a worse approximation". It is better for this paper to provide more empirical results to support this claim. Even thought this paper provides high-level thoughts in illustrating the approximation relating to the rank of a matrix, it is not clear whether the approximation is more/less accurate in practice (no quantitatively indicators provided). Another questions are whether more accurate approximation has better performances (e.g., better optimization performance, or better generalization performance). It is better for this paper to provide these results.
>
> Thank you for this insightful comment. To address it, we added Observation 2, which shows that Ginger provides a more accurate inverse approximation per step than QNG. Regarding the question of whether more accurate approximation leads to better performance, while this remains a nuanced and ongoing topic of discussion [1, 2], our empirical results provide evidence in favor of this connection. As shown in Tables 1 and 2, Ginger consistently improves optimization and generalization across various models and tasks, suggesting that its more accurate approximation translates to real-world benefits.
>
> > In Section 4.3, this paper claims "GINGER is robust to the hyper-parameters". However, from the results shown in Table 4, the training loss is very sensitive to α and \tao. I think this paper should explain it.
>
> Thanks for raising this point. In Table 4, the loss values are on the scale of $10^{-6}$, so the observed differences (e.g., $0.000046$ at most) are practically negligible. We used the term “robust” to mean that the method remains stable and avoids divergence even when hyperparameters are perturbed (a contrast to other second-order methods, which often diverge under such changes [3]). Nonetheless, we agree this term may be misinterpreted, and we have revised the text to clarify our intended meaning.
>
> > It is not clear whether the proposed method is effective than the block-wise FIM approximation(e.g., [1])? [1] Distributed second-order optimization using kronecker-factored approximations, ICLR 2017.
>
> Thanks for the suggestion. Our paper focuses on methods with linear memory complexity to ensure applicability to large-scale models. Block-wise FIM approximations, while more efficient than full FIMs, still have super-linear memory costs. Moreover, these methods depend on specific model architectures. For these reasons, we mention block-wise approaches only in the related work section.
>
> ---
>
> **Summary**
>
> The reviewers raised concerns about unclear statements about the method’s superiority (than QNG) and robustness. We have addressed these by adding Observation 2, clarified our definition of robustness with proper context about the practical scale of variations, and explained our focus on linear memory complexity methods. We appreciate these insights and welcome any further suggestions to strengthen our work.
>
> [1] Bottou et al. "Optimization Methods for Large-Scale Machine Learning." *SIAM Review*, 2018.  \
> [2] Kunstner et al. "Limitations of the empirical Fisher approximation for natural gradient descent." *NeurIPS*, 2019.  \
> [3] Ding et al. "Applying Second Order Optimization to Deep Transformers with Parameter-Efficient Tuning." *ICLR Submission*, 2023.

---

### Review · Reviewer_F2iU · 2025-07-01

**Summary Of Contributions:**

The paper proposes a new approximation of the Gauss-Newton matrix, which is claimed to be more efficient in terms of time and space required for its computation. Asymptotic convergence of the method is shown in the non-convex case.

**Audience:**

Yes

**Claims And Evidence:**

No

**Requested Changes:**

- Please clarify this statement: "this practice does not capture the most useful information in $G_t$, as the optimal low-rank
approximation $Q_tQ_t^\top$ is given by the spectral decomposition or singular value decomposition", clearly state what is the problem with QNG.

- Make this statement more rigorous: "Assumption 4 guarantees the eigenvalues of the moving average in Equation (17) is
upper-bounded, which lower-bounds the eigenvalues of its inverse with some positive number", which moving average are you talking about exactly? Also, this does not seem as straightforward as you make it seem, so you need to write it rigorously. Also, maybe you meant to refer to equation 14 instead of 17.

- Improve the writing, for example, the sentence before equation (6) should say putting or replacing (5) into (4). There are also other instances that need improving, so go through the paper, clean it from such typos and make the writing clearer.

- You should include in your experiments plots that show time and space complexity of your method compared to other methods.

**Strengths And Weaknesses:**

Strengths:

- The construction of the approximation (sec 2.3) seems very interesting.


Weaknesses:

- The paper feels slightly rushed and the writing needs some improvements (correct some typos and improve writing, see  requested changes)

- Some statements are not rigorously justified/ are unclear. For example the statement about QNG needs more elaboration. Also, in the theory section, there is an unclear statement about the eigen values of the inverse of the approximation (see  requested changes for both  statements)

- The theory is very weak, relying on strong noise assumptions (bounded norm instead of bounded variance) and assumptions about $d_t$. Also, the fact that $\lambda_{min}$ is far from zero is not shown rigorously.

---

> ### Author Response · Authors · 2025-07-15
>
> We thank the reviewers for their thoughtful and constructive feedback. We have carefully addressed each point raised and believe the revisions significantly strengthen the paper.
>
> > Please clarify this statement: "this practice does not capture the most useful information in Gt, as the optimal low-rank approximation QtQt⊤ is given by the spectral decomposition or singular value decomposition", clearly state what is the problem with QNG.
>
> Thank you for the suggestion. We have revised the sentence to clarify that QNG does not yield the optimal low-rank approximation of the true FIM $G_t$ in terms of the matrix norm. Specifically, it holds that $\| Q_t Q_t^\top + \alpha^{\min(\tau, t)} I - G \| \ge \| Q Q^\top  + \alpha^{\min(\tau, t)} I - G \|$, where $Q_t$ is the low-rank matrix from Observation 1, and $Q$ denotes the matrix that minimizes this norm. In other words, QNG does not minimize the approximation error in this sense. To better address your question, we have added Observation 2 in the main paper to show that Ginger yields a better per-step inverse approximation than QNG, thus offering a more meaningful comparison.
>
> > Make this statement more rigorous: "Assumption 4 guarantees the eigenvalues of the moving average in Equation (17) is upper-bounded, which lower-bounds the eigenvalues of its inverse with some positive number", which moving average are you talking about exactly? Also, this does not seem as straightforward as you make it seem, so you need to write it rigorously. Also, maybe you meant to refer to equation 14 instead of 17.
>
> Thank you for pointing this out. You are correct. The moving average refers to Equation (14), not Equation (17). We have corrected this in the revision. Additionally, we acknowledged that the original reasoning was too informal. We have reworked the statement and provided a more rigorous argument in the Appendix, under a weaker version of Assumption 4. This version now bounds the spectral norm of the gradient noise covariance, leading to bounded eigenvalues of the matrix in Equation (14), and hence, a lower bound on the eigenvalues of its inverse.
>
> > Improve the writing, for example, the sentence before equation (6) should say putting or replacing (5) into (4). There are also other instances that need improving, so go through the paper, clean it from such typos and make the writing clearer.
>
> We appreciate your careful reading. We have revised the mentioned sentences and carefully edited the manuscript to correct similar writing issues and improve clarity throughout.
>
> > You should include in your experiments plots that show time and space complexity of your method compared to other methods.
>
> Thank you for the suggestion. In our Experiments section, we have reported actual peak memory usage and compute FLOPs for all methods. These directly reflect the practical space and time complexity. We believe these indicators effectively capture the real-world efficiency of our method.
>
> > The theory is very weak, relying on strong noise assumptions (bounded norm instead of bounded variance) and assumptions about dt. Also, the fact that λmin is far from zero is not shown rigorously.
>
> ---
>
> **Summary**
>
> The reviewers raised concerns about theoretical rigor and presentation clarity. We have addressed these by strengthening our theoretical analysis with more realistic assumptions and rigorous proofs, while improving writing clarity and providing comprehensive experimental metrics throughout the manuscript. We remain open to any additional suggestions for further improvement.

---

### Author Response · Authors · 2025-07-15

Dear reviewers,

We sincerely thank all reviewers for their insightful feedback, detailed suggestions, and thoughtful critiques. Your comments have significantly improved the clarity, rigor, and overall quality of our paper. Below we summarize the most important revisions we made in response to your comments:

1.  We replaced the strong bounded-norm noise assumption with a more realistic bounded-variance assumption and substantially revised the convergence proof to provide stronger theoretical guarantees that better connect to our GGN approximation.

2. We clarified why QNG does not yield the optimal low-rank approximation of the Fisher Information Matrix, added Observation 2 demonstrating Ginger's superior per-step inverse approximation.

3. We added comparisons with the diagonal approximation (DNG) method to further strengthen the results drawn from the experiments.

4. We carefully revised the paper for clearer exposition, fixed all reported technical issues, and included missing definitions and matrix identities for better accessibility.

We are grateful for your thoughtful feedback, which we believe lead to substantial improvements in both the theoretical and empirical components of our work. Thank you again for your valuable insights.

Authors

---

### Decision · Action_Editor_ph2e · 2025-08-10

**Recommendation:** Reject

**Audience:**

Yes

**Audience Explanation:**

The topic of the paper is of interest to many optimization researchers as well as practintioners in machine learning.

**Claims And Evidence:**

No

**Claims Explanation:**

The theoretical contribution relies on strong assumptions without clear justification, and the experimental results do not consistently demonstrate the claimed computation and memory gains. In addition, important comparisons are missing, particularly with other Hessian approximations to the GGN.